

# GMD Perspective: the quest to improve the
# evaluation of groundwater representation in
# continental to global scale models
**Tom Gleeson [1,2], Thorsten Wagener [3], Petra Döll [4], Samuel C Zipper [1, 5], Charles West [3], Yoshihide**
**Wada[6], Richard Taylor [7], Bridget Scanlon [8], Rafael Rosolem[3], Shams Rahman[3], Nurudeen Oshinlaja [9],**
**Reed Maxwell [10], Min-Hui Lo [11], Hyungjun Kim [12], Mary Hill [13], Andreas Hartmann [14,3], Graham Fogg [15],**
**James S. Famiglietti [16], Agnès Ducharne [17], Inge de Graaf [18,19], Mark Cuthbert [9,20], Laura Condon [21],**
**Etienne Bresciani [22], Marc F.P. Bierkens [23, 24]**
[1] Department of Civil Engineering, University of Victoria, Canada
[2] School of Earth and Ocean Sciences, University of Victoria
[3] Department of Civil Engineering, University of Bristol, UK & Cabot Institute, University of Bristol, UK.
[4] Institut für Physische Geographie, Goethe-Universität Frankfurt am Main and Senckenberg Leibniz
Biodiversity and Climate Research Centre Frankfurt (SBiK-F), Frankfurt am Main, Germany
[5] Kansas Geological Survey, University of Kansas
[6] International Institute for Applied Systems Analysis, Laxenburg, Austria
[7] Department of Geography, University College London, UK
[8] Bureau of Economic Geology, The University of Texas at Austin, USA
[9] School of Earth and Environmental Sciences & Water Research Institute, Cardiff University, UK
[10] Department of Geology and Geological Engineering, Colorado School of Mines, USA
[11] Department of Atmospheric Sciences, National Taiwan University, Taiwan
[12] Institute of Industrial Science, The University of Tokyo
[13] Department of Geology, University of Kansas, USA
[14] Chair of Hydrological Modeling and Water Resources, University of Freiburg, Germany
[15] Department of Land, Air and Water Resources and Earth and Planetary Sciences, University of
California, Davis, USA
[16] School of Environment and Sustainability and Global Institute for Water Security, University of
Saskatchewan, Saskatoon, Canada
[17] Sorbonne Université, CNRS, EPHE, IPSL, UMR 7619 METIS, Paris, France
[18] Chair or Environmental Hydrological Systems, University of Freiburg, Germany
[19] Water Systems and Global Change Group, Wageningen University, Wageningen, Netherlands
[20] School of Civil and Environmental Engineering, The University of New South Wales, Sydney, Australia
[21] Department of Hydrology & Atmospheric Sciences, University of Arizona, Tucson, Arizona, USA
[22] Center for Advanced Studies in Arid Zones (CEAZA), La Serena, Chile
[23] Physical Geography, Utrecht University, Utrecht, Netherlands
[24] Deltares, Utrecht, Netherlands





# **Abstract**

Continental- to global-scale hydrologic and land surface models increasingly include representations of

the groundwater system. Such large-scale models are essential for examining, communicating, and

understanding the dynamic interactions between the Earth System above and below the land surface as

well as the opportunities and limits of groundwater resources. We argue that both large-scale and

regional-scale groundwater models have utility, strengths and limitations so continued modeling at both

scales is essential and mutually beneficial. A crucial quest is how to evaluate the realism, capabilities and

performance of large-scale groundwater models given their modeling purpose of addressing large-scale

science or sustainability questions as well as limitations in data availability and commensurability.

Evaluation should identify if, when or where large-scale models achieve their purpose or where

opportunities for improvements exists so that such models better achieve their purpose. We suggest

that reproducing the spatio-temporal details of regional-scale models and matching local data is not a

relevant goal. Instead, it is important to decide on reasonable model expectations regarding when a

large scale model is performing 'well enough' in the context of its specific purpose. The decision of

reasonable expectations is necessarily subjective even if the evaluation criteria is quantitative. Our

objective is to provide recommendations for improving the evaluation of groundwater representation in

continental- to global-scale models. We describe current modeling strategies and evaluation practices,

and subsequently discuss the value of three evaluation strategies: 1) comparing model outputs with

available observations of groundwater levels or other state or flux variables (observation-based

evaluation); 2) comparing several models with each other with or without reference to actual

observations (model-based evaluation); and 3) comparing model behavior with expert expectations of

hydrologic behaviors in particular regions or at particular times (expert-based evaluation). Based on

evolving practices in model evaluation as well as innovations in observations, machine learning and

expert elicitation, we argue that combining observation-, model-, and expert-based model evaluation

approaches, while accounting for commensurability issues, may significantly improve the realism of
groundwater representation in large-scale models. Thus advancing our ability for quantification,
understanding, and prediction of crucial Earth science and sustainability problems. We encourage
greater community-level communication and cooperation on this quest, including among global
hydrology and land surface modelers, local to regional hydrogeologists, and hydrologists focused on
model development and evaluation.
**1.  INTRODUCTION: why and how is groundwater modeled at continental to global scales?**
Groundwater is the largest human- and ecosystem-accessible freshwater storage component of the
hydrologic cycle (UNESCO, 1978; Margat & Van der Gun, 2013; Gleeson et al., 2016). Therefore, better
understanding of groundwater dynamics is critical at a time when the 'great acceleration' (Steffen et al.,
2015) of many human-induced processes is increasing stress on water resources (Wagener et al., 2010;
Montanari et al., 2013; Sivapalan et al., 2014; van Loon et al., 2016), especially in regions with limited
data availability and analytical capacity. Groundwater is often considered to be an inherently regional
rather than global resource or system. This is partially reasonable because local to regional peculiarities
of hydrology, politics and culture are paramount to groundwater resource management (Foster et al.
2013) and groundwater dynamics in different continents are less directly connected and coupled than
atmospheric dynamics. Regional-scale analysis and models are essential for addressing local to regional
groundwater issues. Generally, regional scale modeling is a mature, well-established field (Hill &
Tiedeman, 2007; Kresic, 2009; Zhou & Li, 2011; Hiscock & Bense, 2014; Anderson et al. 2015a) with clear
and robust model evaluation guidelines (e.g. ASTM, 2016; Barnett et al., 2012). Regional models have
been developed around the world; for example, Rossman & Zlotnik (2014) and Vergnes et al. (2020)
synthesize regional-scale groundwater models across the western United States and Europe,
respectively.




Yet, important global aspects of groundwater both as a resource and as part of the Earth System are
emerging (Gleeson et al. 2020). First, our increasingly globalized world trades virtual groundwater and
other groundwater-dependent resources in the food-energy-water nexus, and groundwater often
crosses borders in transboundary aquifers. A solely regional approach can be insufficient to analysing
and managing these complex global interlinkages. Second, from an Earth system perspective,
groundwater is part of the hydrological cycle and connected to the atmosphere, oceans and the deeper
lithosphere. A solely regional approach is insufficient to uncover and understand the complex
interactions and teleconnections of groundwater within the Earth System. Regional approaches
generally focus on important aquifers which underlie only a portion of the world's land mass or
population and do not include many other parts of the land surface that may be important for processes
like surface water-groundwater exchange flows and evapotranspiration. A global approach is also
essential to assess the impact of groundwater depletion on sea level rise, since groundwater storage loss
rate on all continents of the Earth must be aggregated. Thus, we argue that groundwater is
simultaneously a local, regional, and increasingly global resource and system and that examining
groundwater problems, solutions, and interactions at all scales is crucial. As a consequence, we urgently
require predictive understanding about how groundwater, used by humans and connected with other
components of the Earth System, operates at a variety of scales.

Based on the arguments above for considering global perspectives on groundwater, we see four specific
purposes of representing groundwater in continental- to global-scale hydrological or land surface
models and their climate modeling frameworks:
(1) To understand and quantify interactions between groundwater and past, present and future
climate. Groundwater systems can have far-reaching effects on climate affecting modulation of



surface energy and water partitioning with a long-term memory (Anyah et al., 2008; Maxwell and
Kollet, 2008; Koirala et al. 2013; Krakauer et al., 2014; Maxwell et al., 2016; Taylor, et al., 2013;
Meixner et et, 2018; Wang et al., 2018; Keune et al., 2018). While there have been significant
advances in understanding the role of lateral groundwater flow on evapotranspiration (Maxwell &
Condon, 2016; Bresciani et al, 2016), the interactions between climate and groundwater over
longer time scales (Cuthbert et al., 2019) as well as between irrigation, groundwater, and climate
(Condon and Maxwell, 2019; Condon et al 2020) remain largely unresolved. Additionally, it is well
established that old groundwater with slow turnover times are common at depth (Befus et al.
2017; Jasechko et al. 2017). Groundwater connections to the atmosphere are well documented in
modeling studies (e.g. Forrester and Maxwell, 2020).  Previous studies have demonstrated
connections between the atmospheric boundary layer and water table depth (e.g. Maxwell et al
2007; Rahman et al, 2015), under land cover disturbance (e.g. Forrester et al 2018), under
extremes (e.g. Kuene et al 2016) and due to groundwater pumping (Gilbert et al 2017).  While a
number of open source platforms have been developed to study these connections (e.g. Maxwell
et al 2011; Shrestha et al 2014; Sulis, 2017) these platforms are regional to continental in extent.
Recent work has shown global impacts of groundwater on atmospheric circulation (Wang et al
2018), but groundwater is still quite simplified in this study.
(2) To understand and quantify two-way interactions between groundwater, the rest of the
hydrologic cycle, and the broader Earth System. As the main storage component of the freshwater
hydrologic cycle, groundwater systems support baseflow levels in streams and rivers, and thereby
ecosystems and agricultural productivity and other ecosystem services in both irrigated and
rainfed systems (Scanlon et al., 2012; Qiu et al., 2019; Visser, 1959; Zipper et al., 2015, 2017).
When pumped groundwater is transferred to  oceans (Konikow 2011; Wada et al., 2012; Döll et
al., 2014a; Wada, 2016; Caceres et al., 2020; Luijendijk et al. 2020), resulting sea-level rise can





impact salinity levels in coastal aquifers, and freshwater and solute inputs to the ocean (Moore,
2010; Sawyer et al., 2016). Difficulties are complicated by international trade of virtual
groundwater which causes aquifer stress in disparate regions (Dalin et al., 2017)
(3) To inform water decisions and policy for large, often transboundary groundwater systems in an
increasingly globalized world (Wada & Heinrich, 2013; Herbert & Döll, 2019). For instance,
groundwater recharge from large-scale models has been used to quantify groundwater resources
in Africa, even though large-scale models do not yet include all recharge processes that are
important in this region (Taylor et al., 2013; Jasechko et al. 2014; Cuthbert et al., 2019; Hartmann
et al., 2017).
(4) To create visualizations and interactive opportunities that inform citizens and consumers, whose
decisions have global-scale impacts, about the state of groundwater all around the world such as
the World Resources Institute's Aqueduct website (https://www.wri.org/aqueduct), a decision-
support tool to identify and evaluate global water risks.
The first two purposes are science-focused while the latter two are sustainability-focused. In sum,
continental- to global-scale hydrologic models incorporating groundwater offer a coherent scientific
framework to examine the dynamic interactions between the Earth System above and below the land
surface, and are compelling tools for conveying the opportunities and limits of groundwater resources
to people so that they can better manage the regions they live in, and better understand the world
around them. We consider both large-scale and regional-scale models to be useful practices that should
both continue to be conducted rather than one replacing another. Ideally large-scale and regional-scale
models should benefit from the other since each has strengths and weaknesses and together the two
practices enrich our understanding and support the management of groundwater across scales (Section

154    2).

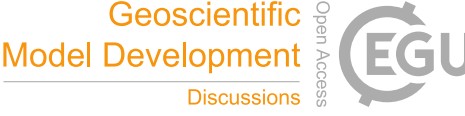

The challenge of incorporating groundwater processes into continental- or global-scale models is
formidable and sometimes controversial. Some of the controversy stems from unanswered questions
about how best to represent groundwater in the models whereas some comes from skepticism about
the feasibility of modelling groundwater at non-traditional scales. We advocate for the representation of
groundwater stores and fluxes in continental to global models for the four reasons described above. We
do not claim to have all the answers on how best to meet this challenge. We contend, however, that the
hydrologic community needs to work deliberately and constructively towards effective representations
of groundwater in global models.

Driven by the increasing recognition of the purpose of representing groundwater in continental- to
global-scale models, many global hydrological models and land surface models have incorporated
groundwater to varying levels of complexity depending on the model provenance and purpose. Different
from regional-scale groundwater models that generally focus on subsurface dynamics, the focus of these
models is on estimating either runoff and streamflow (hydrological models) or land-atmosphere water
and energy exchange (land surface models). Simulation of groundwater storages and hydraulic heads
mainly serve to quantify baseflow that affects streamflow during low flow periods or capillary rise that
increases evapotranspiration. Some land-surface models use approaches based on the topographic
index to simulate fast surface and slow subsurface runoff based on the fraction of saturated area in the
grid cell (Clark et al., 2015; Fan et al., 2019); groundwater in these models does not have water storage
or  hydraulic heads (Famiglietti & Wood, 1994; Koster et al., 2000; Niu et al., 2003; Takata et al., 2003).
In many hydrological models, groundwater is represented as a linear reservoir that is fed by
groundwater recharge and drains to a river in the same grid cell (Müller Schmied et al., 2014; Gascoin et
al., 2009; Ngo-Duc et al., 2007).  Time series of groundwater storage but not hydraulic heads are
computed. This prevents simulation of lateral groundwater flow between grid cells, capillary rise and





two-way exchange flows between surface water bodies and groundwater (Döll et al., 2016). However,
representing groundwater as a water storage compartment that is connected to soil and surface water
bodies by groundwater recharge and baseflow and is affected by groundwater abstractions and returns,
enables global-scale assessment of groundwater resources and stress (Herbert and Döll, 2019) and
groundwater depletion (Döll et al., 2014a; Wada et al., 2014; de Graaf et al., 2014). In some land surface
models, the location of the groundwater table with respect to the land surface is simulated within each
grid cell to enable simulation of capillary rise (Niu et al., 2007) but, as in the case of simulating
groundwater as a linear reservoir, lateral groundwater transport or two-way surface water-groundwater
exchange cannot be simulated with this approach.

Increasingly, models for simulating groundwater flows between all model grid cells in entire countries or
globally have been developed, either as stand-alone models or as part of hydrological models (Vergnes
& Decharme, 2012; Fan et al., 2013; Lemieux et al. 2008; de Graaf et al., 2017; Kollet et al., 2017;
Maxwell et al., 2015; Reinecke et al., 2018, de Graaf et al 2019). The simulation of groundwater in large-
scale models is a nascent and rapidly developing field with significant computational and
parameterization challenges which have led to significant and important efforts to develop and evaluate
individual models. It is important to note that herein 'large-scale models' refer to models that are
laterally extensive across multiple regions (hundreds to thousands of kilometers) and generally include
the upper tens to hundreds of meters of subsurface and have resolutions sometimes as small as ~1 km.
In contrast, 'regional-scale' models (tens to hundreds of kilometers) have long been developed for a
specific region or aquifer and can include greater depths and resolutions, more complex
hydrostratigraphy and are often developed from conceptual models with significant regional knowledge.
Regional-scale models include a diverse range of approaches from stand-alone groundwater models
(i.e., representing surface water and vadose zone processes using boundary conditions such as recharge)



to fully integrated groundwater-surface water models. In the future, large-scale models could be
developed in a number of different directions which we only briefly introduce here to maintain our
primary focus on model evaluation. One important direction is clearer representation of three-
dimensional geology and heterogeneity including karst (Condon et al. in prep) which should be
considered as part of conceptual model development prior to numerical model implementation.

Now that a number of models that represent groundwater at continental to global scales have been
developed and will continue evolving, it is equally important that we advance how we evaluate these
models. To date, large-scale model evaluation has largely focused on individual models and lacked the
rigor of regional-scale model evaluation, with inconsistent practices between models and little
community-level discussion or cooperation. Overall, we have only a partial and piecemeal understanding
of the capabilities and limitations of different approaches to representing groundwater in large-scale
models. Our objective is to provide clear recommendations for evaluating groundwater representation
in continental and global models**.** We focus on model evaluation because this is the heart of model trust
and reproducibility (Hutton et al., 2016) and improved model evaluation will guide how and where it is
most important to focus future model development. We describe current model evaluation practices
(Section 2) and consider diverse and uncertain sources of information, including observations, models
and experts to holistically evaluate the simulation of groundwater-related fluxes, stores and hydraulic
heads (Section 3). We stress the need for an iterative and open-ended process of model improvement
through continuous model evaluation against the different sources of information. We explicitly
contrast the terminology used herein of 'evaluation' and 'comparison' against terminology such as
'calibration' or 'validation' or 'benchmarking', which suggests a modelling process that is at some point
complete. We extend previous commentaries advocating improved hydrologic process representation
and evaluation in large-scale hydrologic models (Clark et al. 2015; Melsen et al. 2016) by adding expert-



elicitation and machine learning for more holistic evaluation. We also consider model objective and
model evaluation across the diverse hydrologic landscapes which can both uncover blindspots in model
development. It is important to note that we do not consider water quality or contamination, even
though water quality or contamination is important for water resources, management and
sustainability, since large-scale water quality models are in their infancy (van Vliet et al., 2019)

We bring together somewhat disparate scientific communities as a step towards greater community-
level cooperation on these challenges, including global hydrology and land surface modelers, local to
regional hydrogeologists, and hydrologists focused on model development and evaluation. We see three
audiences beyond those currently directly involved in large-scale groundwater modeling that we seek to
engage to accelerate model evaluation: 1) regional hydrogeologists who could be reticent about global
models, and yet have crucial knowledge and data that would improve evaluation; 2) data scientists with
expertise in machine learning, artificial intelligence etc. whose methods could be useful in a myriad of
ways; and 3) the multiple Earth Science communities that are currently working towards integrating
groundwater into a diverse range of models so that improved evaluation approaches are built directly
into model development.
**2.   CURRENT MODEL EVALUATION PRACTICES**
Here we provide a brief overview of the synergies and differences between regional-scale and large-
scale model evaluation and development as well as the imitations of current evaluation practices for
large-scale models.

**2.1 Synergies between regional-scale and large-scales**





Regional-scale and large-scale groundwater models are both governed by the same physical equations
and share many of the same challenges.  Like large-scale models, some regional-scale models have
challenges with representing important regional hydrologic processes such as mountain block recharge
(Markovich et al. 2019), and data availability challenges (such as the lack of reliable subsurface
parameterization and hydrologic monitoring data) are common. We propose there are largely untapped
potential synergies between regional-scale and large-scale models based on these commonalities and
the inherent strengths and limitations of each scale (Section 1).

Much can be learned from regional-scale models to inform the development and evaluation of large-
scale groundwater models. Regional-scale models are evaluated using a variety of data types, some of
which are available and already used at the global scale and some of which are not. In general, the most
common data types used for regional-scale groundwater model evaluation match global-scale
groundwater models: hydraulic head and either total streamflow or baseflow estimated using
hydrograph separation approaches (eg. RRCA, 2003; Woolfenden and Nishikawa, 2014; Tolley et al.,
2019). However, numerous data sources unavailable or not currently used at the global scale have also
been applied in regional-scale models, such as elevation of surface water features (Hay et al., 2018),
existing maps of the potentiometric surface (Meriano and Eyles, 2003), and dendrochronology (Schilling
et al., 2014) - these and other 'non-classical' observations (Schilling et al. 2019) could be the inspiration
for model evaluation of large-scale models in the future but are beyond our scope to discuss. Further,
given the smaller domain size of regional-scale models, expert knowledge and local ancillary data
sources can be more directly integrated and automated parameter estimation approaches such as PEST
are tractable (Leaf et al., 2015; Hunt et al., 2013). We directly build upon this practice of integration of
expert knowledge below in Section 3.3.



We  propose that there may also be potential benefits of large-scale models for the development of
regional-scale models. For instance, the boundary conditions of some regional-scale models could be
improved with large-scale model results. The boundary conditions of regional-scale models are often
assumed, calibrated or derived from other models or data. In a regional-scale model, increasing the
model domain (moving the boundary conditions away from region of interests) or incorporating more
hydrologic processes (for example, moving the boundary condition from recharge to the land surface
incorporating evapotranspiration and infiltration) both can reduce the impact of boundary conditions on
the region and problem of interest. Another potential benefit of large-scale models for regional-scale
models is the more fulsome inclusion of large-scale hydrologic and human processes that could further
enhance the ability of regional-scale models to address both the science-focused and sustainability-
focused purposes described in Section 1. For example, the stronger representation of large-scale
atmospheric processes means that the downwind impact of groundwater irrigation on
evapotranspiration on precipitation and streamflow can be assessed (DeAngelis et al., 2010; Kustu et al.,
2011). Or, the effects of climate change and increased water use that affect the inflow of rivers into the
regional modelling domain can be taken from global scale analyses (Wada and Bierkens, 2014 ). Also,
regional groundwater depletion might be largely driven by virtual water trade which can be better
represented in global analysis and models than regional-scale models (Dalin et al. 2017). Therefore the
processes and results of large-scale models could be used to make regional-scale models even more
robust and better address key science and sustainability questions.

Given the strengths of regional models, a potential alternative to development of large-scale
groundwater models would be combining or aggregating multiple regional models in a patchwork
approach (as in Zell and Sanford, 2020) to provide global coverage. This would have the advantage of
better respecting regional differences but potentially create additional challenges because the regional



models would have different conceptual models, governing equations, boundary conditions etc. in
different regions. Some challenges of this patchwork approach include 1) the required collaboration of a
large number of experts from all over the world over a long period of time; 2) regional groundwater flow
models alone are not sufficient, they need to be integrated into a hydrological model so that
groundwater-soil water and the surface water-groundwater interactions can be simulated; 3) the extent
of regional aquifers does not necessarily coincide with the extent of river basins; and 4) the bias of
regional groundwater models towards important aquifers which as described above, underlie only a
portion of the world's land mass or population and may bias estimates of fluxes such as surface water-
groundwater exchange or evapotranspiration. Given these challenges, we argue that a patchwork
approach of integrating multiple regional models is a compelling idea but likely insufficient to achieve
the purposes of large-scale groundwater modeling described in Section 1. Although this nascent idea of
aggregating regional models is beyond the scope of this manuscript, we consider this an important
future research avenue, and encourage further exploration and improvement of regional-scale model
integration from the groundwater modeling community.

**2.2 Differences between regional-scale and large-scales**
Although there are important similarities and potential synergies across scales, it is important to
consider how or if large-scale models are fundamentally different to regional-scale models, especially in
ways that could impact evaluation. The primary differences between large-scale and regional-scale
models are that large-scale models (by definition) cover larger areas and, as a result, typically include
more data-poor areas and are generally built at coarser resolution. These differences impact evaluations
in at least five relevant ways:
1) Commensurability errors (also called 'representativeness' errors) occur either when modelled grid

values are interpolated and compared to an observation 'point' or when aggregation of observed



'point' values are compared to a modelled grid value (Beven, 2005; Tustison et al., 2001; Beven,
2016; Pappenberger et al., 2009; Rajabi et al., 2018). For groundwater models in particular,
commensurability error will depend on the number and locations of observation points, the
variability structure of the variables being compared such as hydraulic head and the interpolation or
aggregation scheme applied (Tustison et al., 2001; Pappenberger et al., 2009; Reinecke et al., 2020).
Commensurability is a problem for most scales of modelling, but likely more significant the coarser
the model. Regional-scale groundwater models typically have fewer (though not insignificant)
commensurability issues due to smaller grid cell sizes compared to large-scale models.
2) Specificity to region, objective and model evaluation criteria because regional-scale models are
developed specifically for a certain region and modeling or management objective whereas large-
scale models are often more general and include different regions. As a result, large-scale models
often have greater heterogeneity of processes and parameters, may not adopt the same calibration
targets and variables, and are not subject to the policy or litigation that sometimes drives model
evaluation of regional-scale models.
3) Computational requirements can be immense for large-scale models which leads to challenges with
uncertainty and sensitivity analysis. While some regional-scale models also have large
computational demands, large-scale models cover larger domains and are therefore more
vulnerable to this potential constraint.
4) Data availability for large-scale models can be limited because they typically include data-poor
areas, which leads to challenges when only using observations for model evaluation. While data
availability also affects regional-scale models, they are often developed for regions with known
hydrological challenges based on existing data and/or modeling efforts are preceded by significant
regional data collection from detailed sources (such as local geological reports) that are not often
included in continental to global datasets used for large-scale model parameterization.





5) Subsurface detail in regional-scale models routinely include heterogeneous and anisotropic
parameterizations which could be improved in future large-scale models. For example, intense
vertical anisotropy routinely induces vertical flow dynamics from vertical head gradients that are
tens to thousands of times greater than horizontal gradients which profoundly alter the meaning of
the deep and shallow groundwater levels, with only the latter remotely resembling the actual water
table. In contrast, currently most large-scale models use a single vertically homogeneous value for
each grid cell, or at best have two layers (de Graaf et al,. 2017)

**2.3 Limitations of current evaluation practices for large-scale models**
Evaluation of large-scale models has often focused on streamflow or evapotranspiration observations
but joint evaluation together with groundwater-specific variables is appropriate and necessary (e.g.
Maxwell et al. 2015; Maxwell and Condon, 2016). Groundwater-specific variables useful for evaluating
the groundwater component of large-scale models include a) hydraulic head or water table depth; b)
groundwater storage and groundwater storage changes which refer to long-term, negative or positive
trends in groundwater storage where long-term, negative trends are called groundwater depletion; c)
groundwater recharge; d) flows between groundwater and surface water bodies; and e) human
groundwater abstractions and return flows to groundwater. It is important to note that groundwater
and surface water hydrology communities often have slightly different definitions of terms like recharge
and baseflow (Barthel, 2014); we therefore suggest trying to precisely define the meanings of such
words using the actual hydrologic fluxes which we do below. Table 1 shows the availability of
observational data for these variables but does not evaluate the quality and robustness of observations.
Overall there are significant inherent challenges of commensurability and measurability of groundwater
observations in the evaluation of large-scale models.  We describe the current model evaluation
practices for each of these variables here:






a) Simulated hydraulic heads or water table depth in large scale models are frequently compared
to well observations, which are often considered the crucial data for groundwater model
evaluation. Hydraulic head observations from a large number groundwater wells (>1 million)
have been used to evaluate the spatial distribution of steady-state heads (Fan et al., 2013, de
Graaf et al., 2015; Maxwell et al., 2015; Reinecke et al., 2019a, 2020). Transient hydraulic heads
with seasonal amplitudes (de Graaf et al. 2017), declining heads in aquifers with groundwater
depletion (de Graaf et al. 2019) and daily transient heads (Tran et al 2020) have also been
compared to well observations. All evaluation with well observations is severely hampered by
the incommensurability of point values of observed head with simulated heads that represent
averages over cells of a size of tens to hundreds square kilometers; within such a large cell, land
surface elevation, which strongly governs hydraulic head, may vary a few hundred meters, and
average observed head strongly depends on the number and location of well within the cell
(Reinecke et al., 2020). Additional concerns with head observations are the 1) strong sampling
bias of wells towards accessible locations, low elevations, shallow water tables, and more
transmissive aquifers in wealthy, generally temperate countries (Fan et al., 2019); 2) the impacts
of pumping which may or may not be well known; 3) observational errors and uncertainty (Post
and von Asmuth, 2013; Fan et al., 2019); and 4) that heads can reflect the poro-elastic effects of
mass loading and unloading rather than necessarily aquifer recharge and drainage (Burgess et al,
2017). To date, simulated hydraulic heads have more often been compared to observed heads
(rather than water table depth) which results in lower relative errors (Reinecke et al., 2020)
because the range of heads (10s to 1000s m head) is much larger than the range of water table
depths (<1 m to 100s m).





b) Simulated groundwater storage trends or anomalies in large-scale hydrological models have been evaluated using observations of groundwater well levels combined with estimates of storage parameters, such as specific yield; local-scale groundwater modeling; and translation of regional total water storage trends and anomalies from satellite gravimetry (GRACE: Gravity Recovery And Climate Experiment) to groundwater storage changes by estimating changes in other hydrological storages (Döll et al., 2012; 2014a). Groundwater storage changes volumes and rates have been calculated for numerous aquifers, primarily in the United States, using calibrated groundwater models, analytical approaches, or volumetric budget analyses (Konikow, 2010). Regional-scale models have also been used to simulate groundwater storage trends untangling the impacts of water management during drought (Thatch et al. 2020). Satellite gravimetry (GRACE) is important but has limitations (Alley and Konikow, 2015). First, monthly time series of very coarse-resolution groundwater storage are indirectly estimated from observations of total water storage anomalies by satellite gravimetry (GRACE) but only after model- or observation-based subtraction of water storage changes in glaciers, snow, soil and surface water bodies (Lo et al., 2016; Rodell et al., 2009; Wada, 2016). As soil moisture, river or snow dynamics often dominate total water storage dynamics, the derived groundwater storage dynamics can be so uncertain that severe groundwater drought cannot be detected in this way (Van Loon et al., 2017). Second, GRACE cannot detect the impact of groundwater abstractions on groundwater storage unless groundwater depletion occurs (Döll et al., 2014a,b). Third, the very coarse resolution can lead to incommensurability but in the opposite direction of well observations. It is important to note that the focus is on storage trends or anomalies since total groundwater storage to a specific depth (Gleeson et al., 2016) or in an aquifer (Konikow, 2010) can be estimated but the total groundwater storage in a specific region or cell cannot be simulated or observed unless the depth of interest is specified (Condon et al., 2020).



c)  Simulated large-scale groundwater recharge (vertical flux across the water table) has been

evaluated using compilations of point estimates of groundwater recharge, results of regional-

scale models, baseflow indices, and expert opinion (Döll and Fiedler, 2008; Hartmann et al.,

2015) or compared between models (e.g. Wada et al. 2010). In general, groundwater recharge is

not directly measurable except by meter-scale lysimeters (Scanlon et al., 2002), and many

groundwater recharge methods such as water table fluctuations and chloride mass balance also

suffer from similar commensurability issues as water table depth data. Although sometimes an

input or boundary condition to regional-scale models, recharge in many large-scale groundwater

models is simulated and thus can be evaluated.

428    d)  The flows between groundwater and surface water bodies (rivers, lakes, wetlands) are

simulated by many models but are generally not evaluated directly against observations of such

flows since they are very rare and challenging. Baseflow (the slowly varying portion of

streamflow originating from groundwater or other delayed sources) or streamflow 'low flows'

(when groundwater or other delayed sources predominate), generally cannot be used to directly

quantify the flows between groundwater and surface water bodies at large scales. Groundwater

discharge to rivers can be estimated from streamflow observations only in the very dense gauge

network and/or if streamflow during low flow periods is mainly caused by groundwater

discharge and not by water storage in upstream lakes, reservoirs or wetlands. These conditions

are rarely met in case of streamflow gauges with large upstream areas that can be used for

comparison to large-scale model output. de Graaf et al. (2019) compared the simulated timing

of changes in groundwater discharge to observations and regional-scale models, but only

compared the fluxes directly between the global- and regional-scale models.  Due to the



challenges of directly observing the flows between groundwater and surface water bodies at
large scales, this is not included in the available data in Table 1; instead in Section 3 we highlight
the potential for using baseflow or the spatial distribution of perennial, intermittent and
ephemeral streams in the future.

e)    Groundwater abstractions have been evaluated by comparison to national, state and county
scale statistics in the U.S. (Wada et al. 2010, Döll et al., 2012, 2014a, de Graaf et al. 2014).
Irrigation is the dominant groundwater use sector in many regions; however, irrigation pumpage
is generally estimated from crop water demand and rarely metered although GRACE and other
remote sensing data have been used to estimate the irrigation water demand (Anderson et al.
2015b). The lack of records or observations of abstraction introduces significant uncertainties
into large-scale models and is simulated and thus can be evaluated. Human groundwater
abstractions and return flows as well as groundwater recharge and the flows between
groundwater and surface water bodies are necessary to simulate storage trends (described
above). But each of these are considered separate observations since they each have different
data sources and assumptions. Groundwater abstraction data at the well scale are severely
hampered by the incommensurability like hydraulic head and recharge described above.
**3. HOW TO IMPROVE THE EVALUATION OF LARGE-SCALE GROUNDWATER MODELS**
Based on Section 2, we argue that the current model evaluation practices are insufficient to robustly
evaluate large-scale models. We therefore propose evaluating large-scale models using at least three
strategies (pie-shapes in Figure 1): observation-, model-, and expert-driven evaluation which are
potentially mutually beneficial because each strategy has its strengths and weaknesses. We are not
proposing a brand new evaluation method here but rather separating strategies to consider the problem



of large-scale model evaluation from different but highly interconnected perspectives. All three
strategies work together for the common goal of 'improved model large-scale model evaluation' which
is what is the centre of Figure 1.

When evaluating large-scale models, it is necessary to first consider reasonable expectations or how to
know a model is 'well enough'. Reasonable expectations should be based on the modeling purpose,
hydrologic process understanding and the plausibly achievable degree of model realism. First, model
evaluation should be clearly linked to the four science- or sustainability-focused purposes of
representing groundwater in large-scale models (Section 1) and second, to our understanding of
relevant hydrologic processes. The objective of large-scale models cannot be to reproduce the spatio-
temporal details that regional-scale models can reproduce. Determining the reasonable expectations is
necessarily subjective, but can be approached using observation-, model-, and expert-driven evaluation.
As a simple first step in setting realistic expectations, we propose that three physical variables can be
used to form more convincing arguments that a large-scale model is well enough:  change in
groundwater storage, water table depth, and regional fluxes between groundwater and surface water.
Below we explore in more detail additional variables and approaches that can support this simple
approach.

Across all three model evaluation strategies of observation-, model-, and expert-driven evaluation, we
advocate three principles underpinning model evaluation (base of Figure 1), none of which we are the
first to suggest but we highlight here as a reminder: 1) model objectives, such as the groundwater
science or groundwater sustainability objective summarised in Section 1, are important to model
evaluation because they provide the context through which relevance of the evaluation outcome is set;
2) all sources of information (observations, models and experts) are uncertain and this uncertainty



needs to be quantified for robust evaluation; and 3) regional differences are likely important for large-
scale model evaluation - understanding these differences is crucial for the transferability of evaluation
outcomes to other places or times.

We stress that we see the consideration and quantification of uncertainty as an essential need across all
three types of model evaluation we describe below, so we discuss it here rather than with model-driven
model evaluation (Section 3.2) where uncertainty analysis more narrowly defined would often be
discussed. We further note that large-scale models have only been assessed to a very limited degree
with respect to understanding, quantifying, and attributing relevant uncertainties. Expanding computing
power, developing computationally frugal methods for sensitivity and uncertainty analysis, and
potentially employing surrogate models can enable more robust sensitivity and uncertainty analysis
such as used in regional-scale models (Habets et al., 2013; Hill, 2006; Hill & Tiedeman, 2007; Reinecke et
al., 2019b). For now, we suggest applying computationally frugal methods such as the elementary effect
test or local sensitivity analysis (Hill, 2006; Morris, 1991; Saltelli et al., 2000). Such sensitivity and
uncertainty analyses should be applied not only to model parameters and forcings but also to model
structural properties (e.g. boundary conditions, grid resolution, process simplification, etc.) (Wagener
and Pianosi, 2019). This implies that the (independent) quantification of uncertainty in all model
elements (observations, parameters, states, etc.) needs to be improved and better captured in available
metadata.

We advocate for considering regional differences more explicitly in model evaluation since likely no
single model will perform consistently across the diverse hydrologic landscapes of the world (Van
Werkhoven et al., 2008). Considering regional differences in large-scale model evaluation is motivated
by recent model evaluation results and is already starting to be practiced. Two recent sensitivity



analyses of large-scale models reveal how sensitivities to input parameters vary in different regions for
both hydraulic heads and flows between groundwater and surface water (de Graaf et al. 2019; Reinecke
et al., 2020). In mountain regions, large-scale models tend to underestimate steady-state hydraulic
head, possibly due to over-estimated hydraulic conductivity in these regions, which  highlights that
model performance varies in different hydrologic landscapes. (de Graaf et al., 2015; Reinecke et al.
2019b). Additionally, there are significant regional differences in performance with low flows for a
number of large-scale models (Zaherpour et al. 2018) likely because of diverse implementations of
groundwater and baseflow schemes. Large-scale model evaluation practice is starting to shift towards
highlighting regional differences as exemplified by two different studies that explicitly mapped
hydrologic landscapes to enable clearer understanding of regional differences. Reinecke et al. (2019b)
identified global hydrological response units which highlighted the spatially distributed parameter
sensitivities in a computationally expensive model, whereas Hartmann et al. (2017) developed and
evaluated models for karst aquifers in different hydrologic landscapes based on different a priori system
conceptualizations.  Considering regional differences in model evaluation suggests that global models
could in the future consider a patchwork approach of different conceptual models, governing equations,
boundary conditions etc. in different regions. Although beyond the scope of this manuscript, we
consider this an important future research avenue.
**3.1 Observation-based model evaluation**
Observation-based model evaluation is the focus of most current efforts and is important because we
want models to be consistent with real-world observations. Section 2 and Table 1 highlight both the
strengths and limitations of current practices using observations. Despite existing challenges, we foresee
significant opportunities for observation-based model evaluation and do not see data scarcity as a
reason to exclude groundwater in large-scale models or to avoid evaluating these models. It is important



to note that most so-called 'observations' are modeled or derived quantities, and often at the wrong
scale for evaluating large-scale models (Table 1; Beven, 2019). Given the inherent challenges of direct
measurement of groundwater fluxes and stores especially at large scales, herein we consider the word
'observation' loosely as any measurements of physical stores or fluxes that are combined with or filtered
through models for an output. For example, GRACE gravity measurements are combined with model-
based estimates of water storage changes in glaciers, snow, soil and surface water for 'groundwater
storage change observations' or streamflow measurements are filtered through baseflow separation
algorithms for 'baseflow observations'.  The strengths and limitations as well as the data availability and
spatial and temporal attributes of different observations are summarized in Table 1 which we hope will
spur more systematic and comprehensive use of observations.

Here we highlight nine important future priorities for improving evaluation using available observations.
The first five priorities focus on current observations (Table 1) whereas the latter four focus on new
methods or approaches:

1)  Focus on transient observations of the water table depth rather than hydraulic head

observations that are long-term averages or individual times (often following well

drilling). Water table depth are likely more robust evaluation metrics than hydraulic

head because water table depth reveals great discrepancies and is a complex function of

the relationship between hydraulic head and topography that is crucial to predicting

system fluxes (including evapotranspiration and baseflow). Comparing transient

observations and simulations instead of  long-term averages or individual times

incorporates more system dynamics of storage and boundary conditions as temporal

patterns are more important than absolute values (Heudorfer et al. 2019). For regions

with significant groundwater depletion, comparing to declining water tables is a useful



strategy (de Graaf et al. 2019), whereas in aquifers without groundwater depletion,

seasonally varying  water table depths are likely more useful observations (de Graaf et

al. 2017).

2)   Use baseflow, the slowly varying portion of streamflow originating from groundwater or

other delayed sources. Döll and Fiedler (2008) included the baseflow index in evaluating

recharge and baseflow has been used to calibrate the groundwater component of a land

surface model (Lo et al. 2008, 2010). But the baseflow index (BFI), linear and nonlinear

baseflow recession behavior or baseflow fraction (Gnann et al., 2019) have not been

used to evaluate any large-scale model that simulates groundwater flows between all

model grid cells. There are limitations of using BFI and baseflow recession characteristics

to evaluate large-scale models (Table 1). Using baseflow only makes sense when the

baseflow separation algorithm is better than the large-scale model itself, which may not

be the case for some large-scale models and only in time periods that can be assumed

to be dominated by groundwater discharge. Similarly, using recession characteristics is

dependent on an appropriate choice of recession extraction methods. But this remains

available and obvious data derived from streamflow or spring flow observations that has

been under-used to date.

3)   Use the spatial distribution of perennial, intermittent, and ephemeral streams as an

observation, which to our best knowledge has not been done by any large-scale model

evaluation. The transition between perennial and ephemeral streams is an important

system characteristic in groundwater-surface water interactions (Winter et al. 1998), so

we suggest that this might be a revealing evaluation criteria although there are similar

limitations to using baseflow. The results of both quantifying baseflow and mapping

perennial streams depend on the methods applied, they are not useful for quantifying





groundwater-surface water interactions when there is upstream surface water storage,

and they do not directly provide information about fluxes between groundwater and

surface water.

4)   Use data on land subsidence to infer head declines or aquifer properties for regions

where groundwater depletion is the main cause of compaction (Bierkens and Wada,

2019). Lately, remote sensing methods such as GPS, airborne and space borne radar and

lidar are frequently used to infer land subsidence rates (Erban et al., 2014). Also, a

number of studies combine geomechanical modelling (Ortega-Guerrero et al 1999;

Minderhoud et al 2017) and geodetic data to explain the main drivers of land

subsidence. A few papers (e.g. Zhang and Burbey 2016) use a geomechanical model

together with a withdrawal data and geodetic observations to estimate hydraulic and

geomechanical subsoil properties.

5)   Consider using socio-economic data for improving model input. For example, reported

crop yields in areas with predominant groundwater irrigation could be used to evaluate

groundwater abstraction rates. Or using well depth data (Perrone and Jasechko, 2019)

to assess minimum aquifer depths or in coastal regions and deltas, the presence of

deeper fresh groundwater under semi-confining layers.

6)   Derive additional new datasets using meta-analysis and/or geospatial analysis such as

gaining or losing stream reaches (e.g., from interpolated head measurements close to

the streams), springs and groundwater-dependent surface water bodies, or tracers.

Each of these new data sources could in principle be developed from available data

using methods already applied at regional scales but do not currently have an 'off the

shelf' global dataset. For example, some large-scale models have been explicitly

compared with residence time and tracer data (Maxwell et al., 2016) which have also



been recently compiled globally (Gleeson et al., 2016; Jasechko et al., 2017). This could
be an important evaluation tool for large-scale models that are capable of simulating
flow paths, or can be modified to do although a challenge of this approach is the
conservativity of tracers. Future meta-analyses data compilations should report on the
quality of the data and include possible uncertainty ranges as well as the mean
estimates.
7) Use machine learning to identify process representations (e.g. Beven, 2020) or
spatiotemporal patterns, for example of perennial streams, water table depths or
baseflow fluxes, which might not be obvious in multi-dimensional datasets and could be
useful in evaluation. For example, Yang et al. (2019) predicted the state of losing and
gaining streams in New Zealand using random forests. A staggering variety of machine
learning tools are available and their use is nascent yet rapidly expanding in geoscience
and hydrology (Reichstein et al., 2019; Shen, 2018; Shen et al., 2018; Wagener et al.,
2020). While large-scale groundwater models are often considered 'data-poor', it may
seem strange to propose using data-intensive machine learning methods to improve
model evaluation. But some of the data sources are large (e.g over 2 million water level
measurements in Fan et al. 2013 although biased in distribution) whereas other
observations such as evapotranspiration (Jung et al., 2011) and baseflow (Beck et al.
2013) are already interpolated and extrapolated using machine learning. Moving
forwards, it is important to consider commensurability while applying machine learning
in this context.
8) Consider comparing models against hydrologic signatures - indices that provide insight
into the functional behavior of the system under study (Wagener et al., 2007; McMilan,
2020). The direct comparison of simulated and observed variables through statistical



error metrics has at least two downsides. One, the above mentioned unresolved
problem of commensurability, and two, the issue that such error metrics are rather
uninformative in a diagnostic sense - simply knowing the size of an error does not tell
the modeller how the model needs to be improved, only that it does (Yilmaz et al.,
2009). One way to overcome these issues, is to derive hydrologically meaningful
signatures from the original data, such as the signatures derived from transient
groundwater levels by Heudorfer et al. (2019). For example, recharge ratio (defined as
the ratio of groundwater recharge to precipitation) might be hydrologically more
informative than recharge alone (Jasechko et al., 2014) or the water table ratio and
groundwater response time (Cuthbert et al. 2019; Opie et al., 2020) which are spatially-
distributed signatures of groundwater systems dynamics. Such signatures might be used
to assess model consistency (Wagener & Gupta, 2005; Hrachowitz et al.2014) by looking
at the similarity of patterns or spatial trends rather than the size of the aggregated
error, thus reducing the commensurability problem.
9)   Understand and quantify commensurability error issues better so that a fairer
comparison can be made across scales using existing data. As described above,
commensurability errors will depend on the number and locations of observation
points, the variability structure of the variables being compared such as hydraulic head
and the interpolation or aggregation scheme applied. While to some extent we may
appreciate how each of these factors affect commensurability error in theory, in
practice their combined effects are poorly understood and methods to quantify and
reduce commensurability errors for groundwater model purposes remain largely
undeveloped. As such, quantification of commensurability error in (large-scale)
groundwater studies is regularly overlooked as a source of uncertainty because it cannot





be satisfactorily evaluated (Tregoning et al., 2012). Currently, evaluation of simulated
groundwater heads is plagued by, as yet, poorly quantified uncertainties stemming from
commensurability errors and we therefore recommend future studies focus on
developing solutions to this problem. An additional, subtle but important and
unresolved commensurability issue can stem from conceptual models. Different
hydrogeologists examining different scales, data or interpreting geology differently can
produce quite different conceptual models of the same region (Troldborg et al. 2007).
We recommend evaluating models with a broader range of currently available data sources (with
explicit consideration of data uncertainty and regional differences) while also simultaneously working to
derive new data sets. Using data (such as baseflow, land subsidence, or the spatial distribution of
perennial, intermittent, and ephemeral streams) that is more consistent with the scale modelled grid
resolution will hopefully reduce the commensurability challenges. However, data distribution and
commensurability issues will likely still be present, which underscores the importance of the two
following strategies.
**3.2. Model-based model evaluation**
Model-based model evaluation, which includes model intercomparison projects (MIP) and model
sensitivity and uncertainty analysis, can be done with or without explicitly using observations. We
describe both inter-model and inter-scale comparisons which could be leveraged to maximize the
strengths of each of these approaches.

The original MIP concept offers a framework to consistently evaluate and compare models, and
associated model input, structural, and parameter uncertainty under different objectives (e.g., climate
change, model performance, human impacts and developments). Early model intercomparisons of



groundwater models focused on nuclear waste disposal (SKI, 1984). Since the Project for the
Intercomparison of Land-Surface Parameterization Schemes (PILPS; Sellers et al., 1993), the first large-
scale MIP, the land surface modeling community has used MIPs to deepen understanding of land
physical processes and to improve their numerical implementations at various scales from regional (e.g.,
Rhône-aggregation project; Boone et al., 2004) to global (e.g., Global Soil Wetness Project; Dirmeyer,
2011). Two examples of recent model intercomparison efforts illustrate the general MIP objectives and
practice. First, ISIMIP (Schewe et al., 2014; Warszawski et al., 2014) assessed water scarcity at different
levels of global warming. Second, IH-MIP2 (Kollet et al., 2017) used both synthetic domains and an
actual watershed to assess fully-integrated hydrologic models because these cannot be validated easily
by comparison with analytical solutions and uncertainty remains in the attribution of hydrologic
responses to model structural errors. Model comparisons have revealed differences, but it is often
unclear whether these stem from differences in the model structures, differences in how the
parameters were estimated, or from other modelling choices (Duan et al., 2006). Attempts for modular
modelling frameworks to enable comparisons (Wagener et al., 2001; Leavesley et al., 2002; Clark et al.,
2008; Fenicia et al., 2011; Clark et al., 2015) or at least shared explicit modelling protocols and boundary
conditions (Refsgaard et al., 2007; Ceola et al., 2015; Warszawski et al., 2014) have been proposed to
reduce these problems.

Inter-scale model comparison - for example, comparing a global model to a regional-scale model - is a
potentially useful approach which is emerging for surface hydrology models (Hattermann et al., 2017;
Huang et al., 2017) and could be applied to large-scale models with groundwater representation. For
example, declining heads and decreasing groundwater discharge have been compared between a
calibrated regional-scale model (RRCA, 2003) and a global model (de Graaf et al., 2019). A challenge to
inter-scale comparisons is that regional-scale models often have more spatially complex subsurface





parameterizations because they have access to local data which can complicate model inter-
comparison. Another approach which may be useful is running large-scale models over smaller
(regional) domains at a higher spatial resolution (same as a regional-scale model) so that model
structure influences the comparison less. In the future, various variables that are hard to directly
observe at large scales but routinely simulated in regional-scale models such as baseflow or recharge
could be used to evaluate large-scale models. In this way, the output fluxes and intermediate spatial
scale of regional models provide a bridge across the "river of incommensurability" between highly
location-specific data such as well observations and the coarse resolution of large-scale models. It is
important to consider that regional-scale models are not necessarily or inherently more accurate than
large-scale models since problems may arise from conceptualization, groundwater-surface water
interactions, scaling issues, parameterization etc.

In order for a regional-scale model to provide a useful evaluation of a large-scale model, there are
several important documentation and quality characteristics it should meet. At a bare minimum, the
regional-scale model must be accessible and therefore meet basic replicability requirements including
open and transparent input and output data and model code to allow large-scale modelers to run the
model and interpret its output. Documentation through peer review, either through a scientific journal
or agency such as the US Geological Survey, would be ideal. It is particularly important that the
documentation discusses limitations, assumptions and uncertainties in the regional-scale model so that
a large-scale modeler can be aware of potential weaknesses and guide their comparison accordingly.
Second, the boundary conditions and/or parameters being evaluated need to be reasonably comparable
between the regional- and large-scale models. For example, if the regional-scale model includes human
impacts through groundwater pumping while the large-scale model does not, a comparison of baseflow
between the two models may not be appropriate. Similarly, there needs to be consistency in the time



period simulated between the two models. Finally, as with data-driven model evaluation, the purpose of
the large-scale model needs to be consistent with the model-based evaluation; matching the hydraulic
head of a regional-scale model, for instance, does not indicate that estimates of stream-aquifer
exchange are valid. Ideally, we recommend developing a community database of regional-scale models
that meet this criteria. It is important to note that Rossman & Zlotnik (2014) review 88 regional-scale
models while a good example of such a repository is the California Groundwater Model Archive
(https://ca.water.usgs.gov/sustainable-groundwater-management/california-groundwater-
modeling.html).

In addition to evaluating whether models are similar in terms of their outputs, e.g. whether they
simulate similar groundwater head dynamics, it is also relevant to understand whether the influence of
controlling parameters are similar across models. This type of analysis provides insights into process
controls as well as dominant uncertainties.  Sensitivity analysis provides the mathematical tools to
perform this type of model evaluation (Saltelli et al., 2008; Pianosi et al., 2016; Borgonovo et al., 2017).
Recent applications of sensitivity analysis to understand modelled controls on groundwater related
processes include the study by Reinecke et al. (2019b) trying to understand parametric controls on
groundwater heads and flows within a global groundwater model. Maples et al. (2020) demonstrated
that parametric controls on groundwater recharge can be assessed for complex models, though over a
smaller domain. As highlighted by both of these studies, more work is needed to understand how to
best use sensitivity analysis methods to assess computationally expensive, spatially distributed and
complex groundwater models across large domains (Hill et al., 2016). In the future, it would be useful to
go beyond parameter uncertainty analysis (e.g. Reinecke et al. 2019b) to begin to look at all of the
modelling decisions holistically such as the forcing data (Weiland et al., 2015) and digital elevation
models (Hawker et al., 2018). Addressing this problem requires advancements in statistics (more



efficient sensitivity analysis methods), computing (more effective model execution), and access to large-
scale models codes (Hutton et al. 2016), but also better utilization of process understanding, for
example to create process-based groups of parameters which reduces the complexity of the sensitivity
analysis study (e.g. Hartmann et al., 2015; Reinecke et al., 2019b).
3.3 Expert-based model evaluation
A path much less traveled is expert-based model evaluation which would develop hypotheses of
phenomena (and related behaviors, patterns or signatures) we expect to emerge from large-scale
groundwater systems based on expert knowledge, intuition, or experience. In essence, this model
evaluation approach flips the traditional scientific method around by using hypotheses to test the
simulation of emergent processes from large-scale models, rather than using large-scale models to test
our hypotheses about environmental phenomena. This might be an important path forward for regions
where available data is very sparse or unreliable. The recent discussion by Fan et al. (2019) shows how
hypotheses about large-scale behavior might be derived from expert knowledge gained through the
study of smaller scale systems such as critical zone observatories. While there has been much effort to
improve our ability to make hydrologic predictions in ungauged locations through the regionalization of
hydrologic variables or of model parameters (Bloeschl et al., 2013), there has been much less effort to
directly derive expectations of hydrologic behavior based on our perception of the systems under study.

Large-scale models could then be evaluated against such hypotheses, thus providing a general
opportunity to advance how we connect hydrologic understanding with large-scale modeling - a strategy
that could also potentially reduce epistemic uncertainty (Beven et al., 2019), and which may be
especially useful for groundwater systems given the data limitations described above. Developing
appropriate and effective hypotheses is crucial and should likely focus on large-scale controlling factors



or relationships between controlling factors and output in different parts of the model domain;
hypotheses that are too specific may only be able to be tested by certain model complexities or in
certain regions. To illustrate the type of hypotheses we are suggesting, we list some examples of
hypotheses drawn from current literature:
● water table depth and lateral flow strongly affect transpiration partitioning (Famiglietti and

Wood, 1994; Salvucci and Entekhabi, 1995; Maxwell & Condon, 2016);

● the percentage of inter-basinal regional groundwater flow increases with aridity or decreases

with frequency of perennial streams (Gleeson & Manning, 2008; Goderniaux et al, 2013; Schaller

and Fan, 2008); or

● human water use systematically redistributes water resources at the continental scale via non-

local atmospheric feedbacks (Al-Yaari et al., 2019; Keune et al., 2018).

Alternatively, it might be helpful to also include hypotheses that have been shown to be incorrect since
models should also not show relationships that have been shown to not exist in nature. For example of
a hypotheses that has recently been shown to be incorrect is that the baseflow fraction (baseflow
volume/precipitation volume) follows the Budyko curve (Gnann et al. 2019) . As yet another alternative,
hydrologic intuition could form the basis of model experiments, potentially including extreme model
experiments (far from the natural conditions). For example, an experiment that artificially lowers the
water table by decreasing precipitation (or recharge directly) could hypothesize the spatial variability
across a domain regarding how 'the drainage flux will increase and evaporation flux will decrease as the
water table is lowered'. These hypotheses are meant only for illustrative purposes and we hope future
community debate will clarify the most appropriate and effective hypotheses. We believe that the
debate around these hypotheses alone will lead to advance our understanding, or, at least highlight
differences in opinion.



Formal approaches are available to gather the opinions of experts and to integrate them into a joint
result, often called expert elicitation (Aspinall, 2010; Cooke, 1991; O'Hagan, 2019). Expert elicitation
strategies have been used widely to describe the expected behavior of environmental or man-made
systems for which we have insufficient data or knowledge to build models directly. Examples include
aspects of future sea-level rise (Bamber and Aspinall, 2013), tipping points in the Earth system (Lenton
et al., 2018), or the vulnerability of bridges to scour due to flooding (Lamb et al., 2017). In the
groundwater community, expert opinion is already widely used to develop system conceptualizations
and related model structures (Krueger et al., 2012; Rajabi et al., 2018; Refsgaard et al., 2007), or to
define parameter priors (Ross et al., 2009; Doherty and Christensen, 2011; Brunner et al., 2012;
Knowling and Werner, 2016; Rajabi and Ataie-Ashtiani, 2016). The term expert opinion may be
preferable to the term expert knowledge because it emphasizes a preliminary state of knowledge
(Krueger et al., 2012).

A critical benefit of expert elicitation is the opportunity to bring together researchers who have
experienced very different groundwater systems around the world. It is infeasible to expect that a single
person could have gained in-depth experience in modelling groundwater in semi-arid regions, in cold
regions, in tropical regions etc. Being able to bring together different experts who have studied one or a
few of these systems to form a group would certainly create a whole that is bigger than the sum of its
parts. If captured, it would be a tremendous source of knowledge for the evaluation of large-scale
groundwater models. Expert elicitation also has a number of challenges including: 1) formalizing this
knowledge in such a way that it is still usable by third parties that did not attend the expert workshop
itself; and 2) perceived or real differences in perspectives, priorities and backgrounds between regional-
scale and large-scale modelers.



So, while expert opinion and judgment play a role in any scientific investigation (O'Hagan, 2019),
including that of groundwater systems, we rarely use formal strategies to elicit this opinion. It is also less
common to use expert opinion to develop hypotheses about the dynamic behavior of groundwater
systems, rather than just priors on its physical characteristics. Yet, it is intuitive that information about
system behavior can help in evaluating the plausibility of model outputs (and thus of the model itself).
This is what we call expert-based evaluation herein. Expert elicitation is typically done in workshops with
groups of a dozen or so experts (e.g. Lamb et al., 2018). Upscaling such expert elicitation in support of
global modeling would require some web-based strategy and a formalized protocol to engage a
sufficiently large number of people. Contributors could potentially be incentivized to contribute to the
web platform by publishing a data paper with all contributors as co-authors and a secondary analysis
paper with just the core team as coauthors. We recommend the community develop expert elicitation
strategies to identify effective hypotheses that directly link to the relevant large-scale hydrologic
processes of interest.
**4. CONCLUSIONS: towards a holistic evaluation of groundwater representation in large-scale models**
Ideally, all three strategies (observation-based, model-based, expert-based) should be pursued
simultaneously because the strengths of one strategy might further improve others. For example,
expert- or model-based evaluation may highlight and motivate the need for new observations in certain
regions or at new resolutions. Or observation-based model evaluation could highlight and motivate
further model development or lead to refined or additional hypotheses. We thus recommend the
community significantly strengthens efforts to evaluate large-scale models using all three strategies.
Implementing these three model evaluation strategies may require a significant effort from the scientific
community, so we therefore conclude with two tangible community-level initiatives that would be



excellent first steps that can be pursued simultaneously with efforts by individual research groups or
collaborations of multiple research groups.

First, we need to develop a 'Groundwater Modeling Data Portal' that would both facilitate and
accelerate the evaluation of groundwater representation in continental to global scale models (Bierkens,
2015). Existing initiatives such as IGRAC's Global Groundwater Monitoring Network  (https://www.un-
igrac.org/special-project/ggmn-global-groundwater-monitoring-network) and HydroFrame
(www.hydroframe.org), are an important first step but were not designed to improve the evaluation of
large-scale models and the synthesized data remains very heterogeneous - unfortunately, even
groundwater level time series data often remains either hidden or inaccessible for various reasons. This
open and well documented data portal should include:

a)  observations for evaluation (Table 1) as well as derived signatures (Section 3.1);

b)  regional-scale models that meet the standards described above and could facilitate inter-scale

comparison (Section 3.2) and be a first step towards linking regional models (Section 2.1);

c)  Schematizations, conceptual or perceptual models of large-scale models since these are the

basis of computational models; and

859        d)  Hypothesis and other results derived from expert elicitation (Section 3.3).

Meta-data documentation, data tagging, aggregation and services as well as consistent data structures
using well-known formats (netCDF, .csv, .txt) will be critical to developing a useful, dynamic and evolving
community resource. The data portal should be directly linked to harmonized input data such as forcings
(climate, land and water use etc.) and parameters (topography, subsurface parameters etc.), model
codes, and harmonized output data. Where possible, the portal should follow established protocols,
such as the Dublin Core Standards for metadata (https://dublincore.org) and ISIMIP protocols for
harmonizing data and modeling approach, and would ideally be linked to or contained within an existing
disciplinary repository such as HydroShare (https://www.hydroshare.org/) to facilitate discovery,
maintenance, and long-term support. Additionally, an emphasis on model objective, uncertainty and
regional differences as highlighted (Section 3) will be important in developing the data portal. Like
expert-elicitation, contribution to the data portal could be incentivized through co-authorship in data
papers and by providing digital object identifiers (DOIs) to submitted data and models so that they are
citable. By synthesizing and sharing groundwater observations, models, and hypotheses, this portal
would be broadly useful to the hydrogeological community beyond just improving global model
evaluation.

Second, we suggest ISIMIP, or a similar model intercomparison project, could be harnessed as a
platform to improve the evaluation of groundwater representation in continental to global scale models.
For example, in ISIMIP (Warszawski et al., 2014), modelling protocols have been developed with an
international network of climate-impact modellers across different sectors (e.g. water, agriculture,
energy, forestry, marine ecosystems) and spatial scales. Originally, ISIMIP started with multi-model
comparison (model-based model evaluation), with a focus on understanding how model projections
vary across different sectors and different climate change scenarios (ISIMIP Fast Track). However, more
rigorous model evaluation came to attention more recently with ISIMIP2a, and various observation data,
such as river discharge (Global Runoff Data Center), terrestrial water storage (GRACE), and water use
(national statistics), have been used to evaluate historical model simulation (observation-based model
evaluation). To better understand model differences and to quantify the associated uncertainty sources,
ISIMIP2b includes evaluating scenarios (land use, groundwater use, human impacts, etc) and key
assumptions (no explicit groundwater representation, groundwater availability for the future, water
allocation between surface water and groundwater), highlighting that different types of hypothesis
derived as part of the expert-based model evaluation could possibly be simulated as part of the ISIMIP



process in the future. While there has been a significant amount of research and publications on MIPs
including surface water availability, limited multi-model assessments for large-scale groundwater
studies exist. Important aspects of MIPs in general could facilitate all three model evaluation strategies:
community-building and cooperation with various scientific communities and research groups, and
making the model input and output publicly available in a standardized format.

Large-scale hydrologic and land surface models increasingly represent groundwater, which we envision
will lead to a better understanding of large-scale water systems and to more sustainable water resource
use. We call on various scientific communities to join us in this effort to improve the evaluation of
groundwater in continental to global models. As described by examples above, we have already started
this journey and we hope this will lead to better outcomes especially for the goals of including
groundwater in large-scale models that we started with above: improving our understanding of Earth
system processes; and informing water decisions and policy. Along with the community currently
directly involved in large-scale groundwater modeling, above we have made pointers to other
communities who we hope will engage to accelerate model evaluation: 1) regional hydrogeologists, who
would be useful especially in expert-based model evaluation (Section 3.3); 2) data scientists with
expertise in machine learning, artificial intelligence etc. whose methods could be useful especially for
observation- and model-based model evaluation (Sections 3.1 and 3.2); and 3) the multiple Earth
Science communities that are currently working towards integrating groundwater into a diverse range of
models so that improved evaluation approaches are built directly into model development. Together we
can better understand what has always been beneath our feet, but often forgotten or neglected.





**Competing interests:** The authors declare that they have no conflict of interest.

**Acknowledgements:**
The commentary is based on a workshop at the University of Bristol and significant debate and
discussion before and after. This community project was directly supported by a Benjamin Meaker
Visiting Professorship at the Bristol University to TG and a Royal Society Wolfson Award to TW
(WM170042). We thank many members of the community who contributed to the discussions,
especially at the IGEM (Impact of Groundwater in Earth System Models) workshop in Taiwan.

**Author Contributions**: (using the CRediT taxonomy which offers standardized descriptions of author
contributions) conceptualization and writing original draft: TG, TW and PD; writing - review and
editing:all co-authors. Authors are ordered by contribution for the first three coauthors (TG, TW and PD)
and then ordered in reverse alphabetical order for all remaining coauthors.

**Code and data availability:** This Perspective paper does not present any computational results. There is
therefore no code or data associated with this paper.









**Table 1. Available observations for evaluating the groundwater component of large-scale models**

| Data type | Strengths | Limitations | Data availability and spatial resolution |
|---|---|---|---|
| **Available observations already used to evaluate large-scale models** | | | |
| Hydraulic heads or water table depth (averages or single times) | Direct observation of groundwater levels and storage | observations biased towards North America and Europe; non- commensurable with large-scale models; mixture of observation times | IGRAC Global Groundwater Monitoring Network; Fan et al., 2013; USGS Point measurements at existing wells |
| Hydraulic heads or water table depth (transient) | Direct observation of changing groundwater levels and storage | As above | time-series available in a few regions, especially through USGS and European Groundwater Drought Initiative Point measurements at existing wells |
| Total water storage anomalies (GRACE) | Globally available and regionally integrated signal of water storage trends and anomalies | Groundwater changes are uncertain model remainder; very coarse spatial resolution and limited period | Various mascons gridded with resolution of ~100,000 km² (Scanlon et al. 2016) which are then processed as groundwater storage change |
| Storage change (regional aquifers) | Regionally integrated response of aquifer | Bias towards North America and Europe | Konikow 2011 Döll et al., 2014a Regional aquifers (10,000s to 100,000s km²) |
| Recharge | Direct inflow of groundwater system | Challenging to measure and upscale | Döll and Fiedler, 2008; Hartmann et al. 2017; Mohan et al. 2018; Moeck et al. 2020 Point to small basin |
| Abstractions | Crucial for groundwater depletion and sustainability studies | National scale data highly variable in quality; downscaling uncertain | de Graaf et al. 2014 Döll et al. 2014 National-scale data down-scaled to grid |
| Streamflow or spring flow observations | Widely available at various scales; low flows can be related to groundwater | Challenging to quantify the flows between groundwater and surface water from streamflow | Global Runoff Data Centre (GRDC) or other data sources; large to small basin; Olarinoye et al. 2020 point measurements of spring flow |



| Evapotranspiration | Widely available; related to groundwater recharge or discharge (for shallow water tables) | Not a direct groundwater observations | Various datasets (Miralles et al., 2016); gridded |
|---|---|---|---|
| **Available observations not being used to evaluate large-scale models** | | | |
| Baseflow index (BFI) or (non-)linear baseflow recession behavior | Possible integrator of groundwater contribution to streamflow over a basin | BFI and k values vary with method; baseflow may be dominated by upstream surface water storage rather than groundwater inflow; can not identify losing river conditions | Beck et al. (2013) Point observations extrapolated by machine learning |
| Perennial stream map | Ephemeral streams are losing streams, whereas perennial streams could be gaining (or impacted by upstream surface water storage) | Mapping perennial streams requires arbitrary streamflow and duration cutoffs; not all perennial streams reaches are groundwater-influenced; does not provide information about magnitude of inflows/outflows. | Schneider et al. (2017) Cuthbert et al. (2019); Spatially continuous along stream networks |
| Gaining or losing stream reaches | Multiple techniques for measurement (interpolated head measurements, streamflow data, water chemistry). Constrains direction of fluxes at groundwater system boundaries | Relevant processes occur at sub-grid-cell resolution. | Not globally available but see Bresciani et al. (2018) for a regional example; Spatially continuous along stream networks |
| Springs and groundwater-dependent surface water bodies | Constrains direction of fluxes at groundwater system boundaries | Relevant processes occur at sub-grid-cell resolution. | Springs available for various regions (e.g. Springer, & Stevens, 2009) but not globally; Point measurements at water feature locations |
| Tracers (heat, isotopes or other geochemical) | Provides information about temporal aspects of groundwater systems (e.g. residence time) | No large-scale models simulate transport processes (Table S1) | Isotopic data compiled (Gleeson et al., 2016; Jasechko et al., 2017) but no global data for heat or other chemistry; Point measurements at existing wells or surface water features |

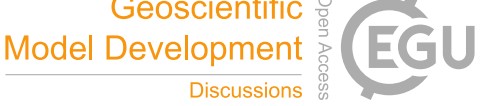

| Surface elevation data (leveling, GPS, radar/lidar) an in particular land subsidence observations | Provides information about changes in surface elevation that are related to groundwater head variations or groundwater head decline | Provides indirect information and needs a geomechanical model to translate to head. Introduces additional uncertainty of geomechnical properties. | Leveling data, GPS data and lidar observations mostly limited to areas of active subsidence (e.g. Minderhoud et al., 2019,2020) and not always open. Global data on elevation change are available from the Sentinel 1 mission. |
|---|---|---|---|


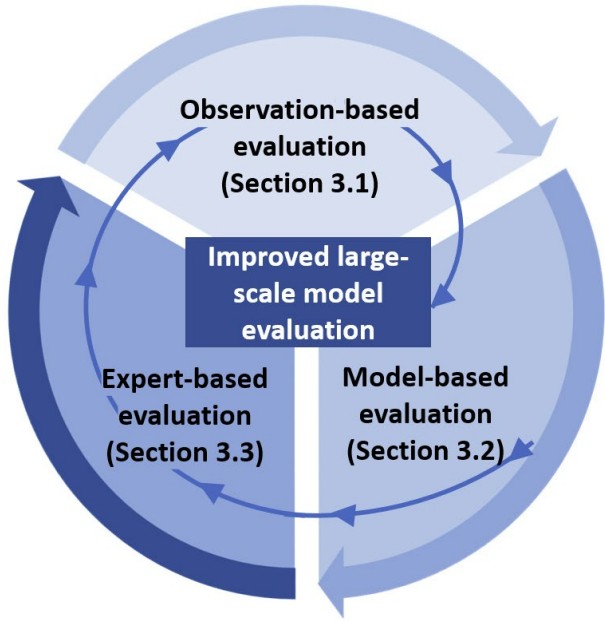


**Figure 1: Improved large-scale model evaluation rests on three pillars: observation-, model-, and**
**expert-based model evaluation. We argue that each pillar is an essential strategy so that all three**
**should be simultaneously pursued by the scientific community.  The three pillars of model evaluation**
**all rest on three core principles related to 1) model objectives, 2) uncertainty and 3) regional**
**differences.**





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
