# Peer review of "GMD Perspective: the quest to improve the"

_Geoscientific Model Development, 2021_

## Author Comment (AC1)

Dear Chief-executive editor David Ham and Topical Editor Jatin Kala

Thank you for handling the reviews of our proposed GMD Perspective article entitled "Improving the evaluation of groundwater modeled at continental to global scales".

The two reviewers are convergent that the manuscript is 'a very well written paper and authors provide important insights about how to move the community forward' and 'very thorough and exhaustive'.

The most significant suggested revision is to add 'examples of existing global scale groundwater models' or 'add a table that summarizes the main characteristics of the existing models at the continental and global scale'. Based on these suggestions, the most significant change to the manuscript we made is adding a new sub-section called "2.1 Brief overview of current large-scale groundwater models" and a new Table 1 which summarizes the main characteristics of existing models at the continental and global scale.

We also address each of the minor comments as we outline below in blue.

Warmly, Tom Gleeson, Thorsten Wagener and Petra Döll

**Anonymous Referee #1**

Authors also discussed challenges in evaluating global groundwater models. For example, the issue of mismatch between the scale of observations and model grid cell exist for any distributed hydrologic model and it is not unique to global groundwater models. The evaluation strategies for global scale models are also similar to any hydrologic models. Therefore, I wonder whether authors could spend more time bringing various view points for model development rather than evaluation as this is the most important challenge in the literature. It would be useful to bring examples of existing global scale groundwater models and discuss their performance to convey the status of science to the readers.

Thank you for this suggestion which we considered in detail. We believe that model evaluation is a key step in the (typically iterative) model development process. Our paper argues that more and other information is available for this step than used so far, thus highlighting how we might advance model development through better model evaluation.

We have added a new sub-section called "2.1 Brief overview of current large-scale groundwater models" and a new Table 1 which summarizes the main characteristics of existing models at the continental and global scale.

Line 173 – This statement is not very clear. Perhaps add "Explicit" to water storage or hydraulic head since these models consider subsurface storage and distribute estimated average water table depth across the domain based on the topographic wetness index.

Good suggestion - we made this change.

Lines 257-271- What about the use of stable and radiogenic isotopes for determining water sources and residence times?

Modified as suggested.

Lines 705-707- Authors discuss the use of estimated recharge or other fluxes from regional scale models to assess global scale groundwater models. However, such flux estimates are not often very accurate and contain large uncertainty. How do authors recommend comparing these uncertain datasets together?

We agree that these flux estimates are not often very accurate and can contain large uncertainty - we modified this sentence to acknowledge this. Several studies have demonstrated that even rather uncertain observations can be useful in constraining the uncertainty in large scale models by limiting acceptable model behaviour. For example, even relatively uncertain recharge estimates for karstic regions show that most models miss the specific processes that allow for the significant recharges expected in those domains (Hartmann et al., 2015, GMD).

We agree that comparing these uncertain datasets is important so we added this sentence: "In such an evaluation, the uncertainty of flux estimates and scale of aggregation are both important to consider."

Table 1- Authors list groundwater storage observations at point scale for model evaluation. Could authors further clarify the sources of these data.

For 'storage change' observations, Table 2 (which used to be Table 1) describes these at the scale of regional aquifers. Hydraulic heads or depth to water table are described as point measurements if this is what the reviewer is referring to. In this case, we have the source of data already in the table.

What may have been confusing about the entry "Storage change (regional aquifers)" in Table 2 is that this data type is not really an observation but an independent estimate derived by various approaches including regional-scale modeling (see Konikow 2011). We have therefore replaced "Storage change (regional aquifers)" with the clearer "Storage change in regional aquifers (independent estimates derived by various methods)".

Overall this is a very well written paper and authors provide important insights about how to move the community forward. Some of the information throughout the paper could be summarized in tables or conceptual figures to highlight the main points of the paper.

Thank you for the overall positive evaluation. We endeavored to summarize the information in the figure and tables.

**Anonymous Referee #2**

The review is very thorough and exhaustive, and I completely agree with the importance of improving the representation of groundwater in model at all scales and I understand the need to transfer knowledge across scale. The paper is very nicely written and I especially appreciated the discussion starting at line 549 about how critical is to use different types of observations (9 are included here) to better evaluate models.

Thank you for the overall positive evaluation.

The first type of observation mentioned refers to the importance of moving away from using averages and highlight how critical is to use transient of observations representing depth to groundwater. The only suggestion I have is to eventually add a table that summarize the main characteristics of the existing models at the continental and global scale. They are all referenced in the paper but I believe a table would be helpful for readers eventually less familiar with all the ongoing model development.

As described above we have added a new sub-section called "2.1 Brief overview of current large-scale groundwater models" and a new Table 1 which summarizes the main characteristics of existing models at the continental and global scale.

Thank you for the opportunity to review the paper!

Thank you for taking the time to review the manuscript!

Minor comments:

Line 92: what do you mean by "teleconnections"?

The term comes from atmospheric sciences where teleconnections are "the climate links between geographically separated regions" (Encyclopedia of Atmospheric Sciences) or "climate anomalies being related to each other at large distances (typically thousands of kilometers)" (wikipedia).

We mean this term in a more general way to infer that parts of the groundwater system (levels, flows etc) in one region could be related to impacts in other regions via physical processes or trade). So we define in the text we add a more general definition of teleconnection as 'groundwater levels or flows in one region linked to geographically separated regions via physical or socio-economic processes'.

Line 211-213: this sentence is not very clear

Reworded for clarity

Line 339-340: What do you mean by "when only using observations for model evaluation"? I imagine this refers to observations not being used for model development, is that correct?

We added the statement "when only using observations for model evaluation" to foreshadow and motivate the later discussion in "3.2. Model-based model evaluation" and "3.3 Expert-based model evaluation" which go beyond just using observations.

---

## Author Response (AR2)

Dear Topical Editor Jatin Kala and Chief-executive editor David Ham,

Thank you for handling the reviews of our proposed GMD Perspective article entitled "Improving the evaluation of groundwater modeled at continental to global scales".

The reviewer suggests accepting subject to minor revisions. This reviewer had three minor comments that we address below in blue, with relevant changes to Table 1 in the manuscript.

Warmly, Tom Gleeson, Thorsten Wagener and Petra Döll

**Anonymous Referee #1**

Authors have done an excellent job incorporating reviewers' comments. Adding Table 1 and proposing a new classification for large-scale groundwater models added significant value to the manuscript. Thank you for your huge efforts on compiling this information. I have three minor comments regarding Table 1 and I appreciate if authors can clarify these points.

Thank you for taking the time to review our manuscript again and your positive feedback on our revision. We clarify each of these points below.

1. Authors indicated that focused recharge processes in models such as HydroGeosphere and ParFlow are not incorporated. As the two-way exchange fluxes between the river cells and model grids are calculated, I wonder why authors indicated that focused recharge is not represented in these models. Does this relate to not explicitly representing river geometry in these models?

This is useful comment – we changed HydroGeoSphere and Parflow to directly representing focused recharge processes.

2. Models such as ParFlow are very flexible in representing subsurface heterogeneity and elastic storages are directly calculated. I wonder why authors indicate that confined condition is not represented in these models.

This is also a useful suggestion. Parflow is very flexible in this way and could be used to represent confined conditions. In the continental applications to date it has not been parameterized in this way so we changed the confined aquifer cells for Parflow and HydroGeoSphere to 'Potentially represented'.

3. One of the criteria in Table 1 is "Groundwater use" and it seems none of the models incorporate this component. Does groundwater use relate to pumping or groundwater evapotranspiration? Please clarify.

Thanks for this final useful suggestion. By groundwater use we did mean pumping so we have added this note to the bottom of the table "Groundwater use means groundwater pumping rather than via evapotranspiration." In the previously submitted manuscript, PCRGLOB-WB was the only model that directly represented this, but we have added Parflow as representing this based on some recent papers.